

# Aspect extraction on user textual reviews using multi-channel convolutional neural network

Aminu Da'u[1,2] and Naomie Salim[1]

[1] School of Computing, Faculty of Engineering, Universiti Teknologi Malaysia, Skudai, Johor, Malaysia
[2] Department of OTM, Hassan Usman Katsina Polytechnic, Katsina, Nigeria

## ABSTRACT

Aspect extraction is a subtask of sentiment analysis that deals with identifying opinion targets in an opinionated text. Existing approaches to aspect extraction typically rely on using handcrafted features, linear and integrated network architectures. Although these methods can achieve good performances, they are time-consuming and often very complicated. In real-life systems, a simple model with competitive results is generally more effective and preferable over complicated models. In this paper, we present a multichannel convolutional neural network for aspect extraction. The model consists of a deep convolutional neural network with two input channels: a word embedding channel which aims to encode semantic information of the words and a part of speech (POS) tag embedding channel to facilitate the sequential tagging process. To get the vector representation of words, we initialized the word embedding channel and the POS channel using pretrained word2vec and one-hot-vector of POS tags, respectively. Both the word embedding and the POS embedding vectors were fed into the convolutional layer and concatenated to a one-dimensional vector, which is finally pooled and processed using a Softmax function for sequence labeling. We finally conducted a series of experiments using four different datasets. The results indicated better performance compared to the baseline models.

# INTRODUCTION

With the growth of textual information on the web, aspect-based sentiment analysis has been widely studied, thereby attracting much attention in the research community. One of the important subtasks of aspect-based sentiment analysis is aspect extraction, which is simply the act of extracting attributes of an entity about which opinions are expressed (*Liu, 2012*). Aspect extraction can generally be performed using either unsupervised (*Qiu et al., 2011*; *Wang & Wang, 2008*) or supervised methods (*Lafferty, Mccallum & Pereira, 2001*; *Poria, Cambria & Gelbukh, 2016*; *Cambria, 2016*). For many years, the state-of-the-art methods of aspect extraction basically depend on the conditional random fields (CRF) (*Lafferty, Mccallum & Pereira, 2001*), recurrent neural network (RNN) (*Irsoy & Cardie, 2014*), linguistic patterns and syntactic rules (*Qiu et al., 2011*;

Corresponding author
Aminu Da'u,
dauaminu@graduate.utm.my

*Popescu & Etzioni, 2005*). Both of these approaches have their own shortcomings. For example, CRF is typically linear in nature. Thus, it requires a large number of datasets to effectively work. RNNs are generally not effective in predicting word labels or phrases that are determined by the context due to their feedback nature. Syntactic rules and linguistic patterns need to be handcrafted and their accuracy generally depends on the grammatical accuracy of the sentences.

To address the aforementioned issues among others, few approaches have been proposed to exploit deep convolutional neural network (CNN) architectures to improve the performance of the aspect extraction models (*Poria, Cambria & Gelbukh, 2016*; *Xu et al., 2018a*). These models do not usually require predefined features to be manually handpicked; instead, they can automatically learn sophisticated features from their datasets. Generally, words are usually represented in the form of a vector and the extraction of the feature is left to the network. Consequently, words with similar semantics can be mapped using these models to nearby locations in their coordinate systems.

Even though these approaches have shown better performances than their prior approaches, however, there are some important issues worth to be considered for further improvement: First, most of the existing approaches typically used only general pretrained word embeddings such as Google Word2vec or Glove embeddings as the main semantic feature for the aspect extraction, although word embeddings have shown effectiveness in capturing both syntactic and semantic information of words. However, in some cases, due to the distributional hypothesis, word embeddings alone fail to efficiently capture the syntactic information of some aspect terms, for example, in the latent space, bad and good are typically mapped together as neighbors while analyzing these words is very critical in aspect classification. Moreover, due to the complexity of the aspect extraction task, fine-grained embeddings are particularly important to achieve a better performance (*Yin et al., 2016*). Therefore, we urge that using a domain-specific embedding is very crucial for information extraction performance. Thus, in this paper, we exploited both the general and domain-specific embeddings to examine which embeddings are superior over the other.

Additionally, most of the previous CNN based aspect extraction models are either stacked (*Ye, Yan & Luo, 2017*) or integrated with other models such as long short term memory (LSTM) (*Dong, Zhang & Yang, 2017*). These consequently increase the complexity of the model parameters. Although these may improve the model performance, according to *Blumer et al. (1987)*, in real-world applications, a simple model is always preferred and more useful over the complicated model. This is particularly important when a model is used for a real-life situation such as chatbot in which a complex model will retard the inferential performance of the model. Thus, achieving a competitive performance while ensuring a simple architecture without manually crafting features and much complexity is always a crucial direction to explore. This paper proposes to achieve such a goal.

Motivated by the above-mentioned issues, this paper proposes an aspect extraction model based on an multichannel convolutional neural network (MCNN) leveraging two different embedding layers: word embedding and part of speech (POS) tag embedding layer. To achieve a simple architecture while ensuring a competitive performance,

we propose a purely CNN model for sequential labeling. A CNN model which is nonlinear network architecture can fit data more easily with relatively few parameters. The major contributions of the proposed model can be summarized as follows:

1. We introduced an MCNN model for aspect extraction leveraging two different input channels: word embeddings and POS Tag embeddings channel to encode the contextual information and enhance sequential tagging of words, respectively.
2. We investigated the importance of using domain-specific embeddings over the general-purpose word embeddings in aspect extraction.
3. We conducted a series of experiments on the SemEval challenge datasets (*Pontiki & Pavlopoulos, 2014*; *Maria et al., 2015*; *Hercig et al., 2016*) and showed that our approach outperformed the baseline methods with significant gains across all the datasets.

The remainder of the paper is arranged as follows. In sections, "Related Work", "The Proposed Model", "Experimental Study", "Results and Discussion" and "Conclusion and Future Direction".

## RELATED WORK

Aspect extraction as the subtask of aspect-based sentiment analysis has been widely studied by many researchers. One of the earliest studies was conducted by *Hu & Liu (2004)* to propose a rule-based method for the explicit aspect categorization. This method was later improved by many approaches among which include the work of *Popescu & Etzioni (2005)* who used point-wise mutual information between the product class and noun phrase for product feature extraction.

Generally, aspect extraction can be performed using either unsupervised (*Qiu et al., 2011*; *Wang & Wang, 2008*; *Popescu & Etzioni, 2005*; *Chen, Mukherjee & Liu, 2015*) or supervised method (*Lafferty, Mccallum & Pereira, 2001*; *Poria, Cambria & Gelbukh, 2016*; *Cambria, 2016*). Our proposed work particularly focuses on the supervised methods which treat aspect extraction task as a sequence labeling problem. The traditional supervised methods are mainly based on hidden Marcok model (*Jin & Ho, 2009*) and CRF (*Lafferty, Mccallum & Pereira, 2001*). With the recent success of deep learning in different areas such as image classification and pattern recognition, several approaches have been proposed to exploit deep learning methods for the aspect extraction. For instance, *Wang et al. (2015)* employed a restricted Boltzmann machine model to jointly address the problem of sentiment–aspect extraction. *Irsoy & Cardie (2014)* utilized RNN and demonstrated its superior performance over the CRF-based models for aspect extraction. This method was later improved by *Pengfei, Shafiq & Helen (2015)*. They applied more sophisticated variants of the RNN using fine-tuned word vectors and additional linguistic features for better improvement. To tag each word with non-aspect or aspect label, a multilayer CNN was proposed by *Poria, Cambria & Gelbukh (2016)*. The authors used syntactic and linguistic patterns to improve the accuracy of the model.

For further improvement, the attention-based model has been used for aspect extraction to learn the representation of the informative words in text review (*Chen et al., 2017*; *Wang, Pan & Dahlmeier, 2017*; *Maria et al., 2015*; *Hercig et al., 2016*). Tree-based methods have been shown effective for improving the performance of the aspect extraction model. For instance, *Yin et al. (2016)* introduced a dependency path approach in which both the dependency and linear contextual information are considered for the word representation. A similar method was proposed by *Wang et al. (2016)* to exploit dependency tree and CRF for better coextraction of aspect and opinion terms. *Xu et al. (2017)* and *Li & Lam (2018)* also exploited deep learning for coextraction of the aspect and opinion terms. Recently, a tree-based CNN was introduced by *Ye, Yan & Luo (2017)*. They applied tree-based convolution over a sentence dependency parse tree. *Luo et al. (2018)* proposed an end-to-end method to integrate BiLSTM, CRF and word embeddings for aspect term extraction.

Our approach is closely related to the work of *Xu et al. (2018b)* in which a double embeddings method has been used to model aspect extraction using two different in-domain word embeddings. However, this method has a drawback in that it solely relies on the word embedding as the main feature and ignore to utilize the POS tag for the sequential tagging. In our approach, POS tags features are utilized in addition to the word embedding features to improve the model performance. Furthermore, unlike the previous methods, we specifically used two different channels as the input to the convolutional network architecture. We used both general and domain-specific embedding in the first channel specifically to capture the syntactic and semantic information of the word, and POS tag embedding in the second channel to specifically improve the sequential labeling of the aspects. To the best of our knowledge, this is the first work to use a MCNN architecture leveraging both word embeddings and POS tag embeddings in different channels for better performance of the aspect extraction model.

## OUR MODEL

Figure 1 illustrates the proposed MCNN architecture. The model is based on the CNN structure proposed in *Kim (2014)*. Specifically, the proposed model is made up of two input layers: word embedding and POS embedding layer. It consists of two convolutional layers followed by a max pooling layer, rectified linear unit optimizer, a fully connected layer, and a Softmax classifier to predict the multiclass labels of the aspects with labelling space, Y = {B, I, O} with "I", "O" and "B" representing Inside, Outside or Beginning of the aspect term, respectively. Detail of the model is described in the following subsections:

### Input channels

The model typically comprises two sets of vectors, each of which is an input channel to the network (*Kim, 2014*). For the word embedding channel, the main idea is to capture the semantic information of the words. For that, we use both general and domain-specific embeddings. Use this correction: For the general embedding, we used a pretrained word embedding trained on 100 billion words of Google corpus (*Mikolov, Yih & Zweig, 2013*), while for the domain-specific embedding, we specifically used a CBOW (continuous

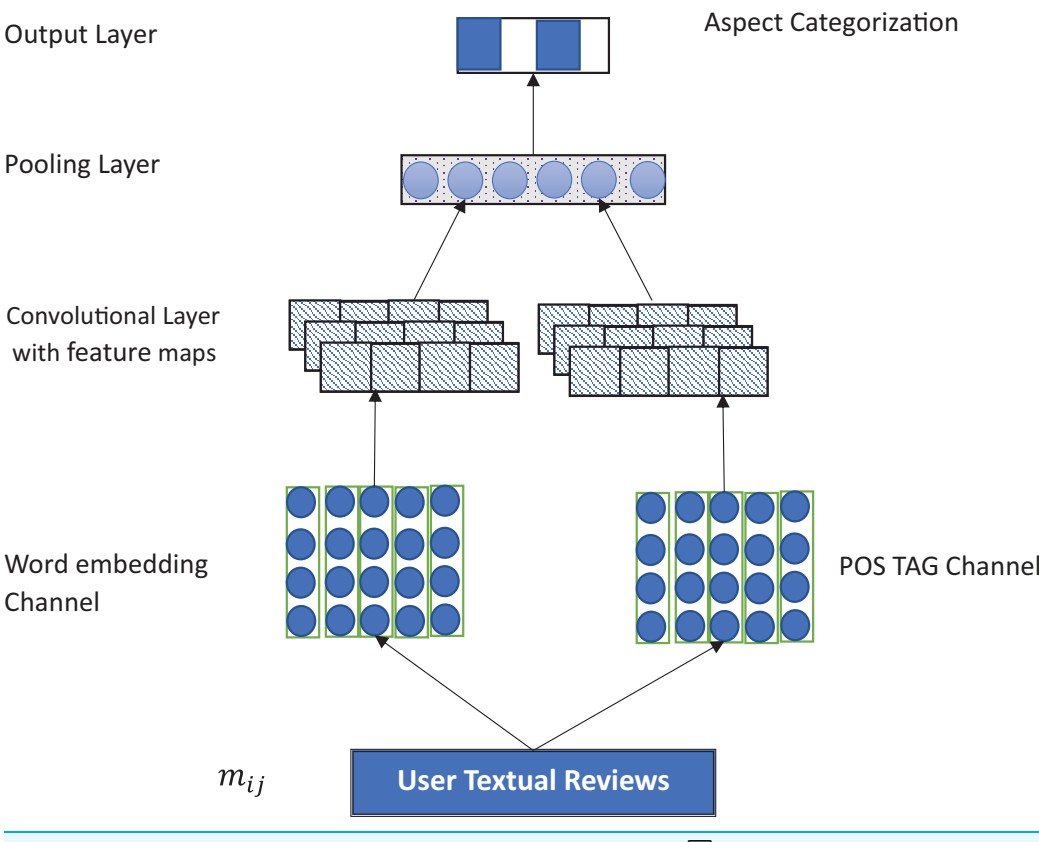

**Figure 1  Overview of the MCNN architecture.**     

bag of words) architecture trained on Amazon and Yelp reviews for the laptop and restaurant domain, respectively. In this case, each word was encoded as 300-dimensional vectors. We use word padding to make sure that all sentences are of the same length. To capture the contextual features of the words, $i$th words are mapped to a $k$-dimensional embedding. The semantic feature of a sentence of length n is given as $|X|_1^n = \{x_1 \ldots \ldots, x_n\}, X \in R^K$.

For the POS Tag embeddings, the main idea is to improve the aspect extraction process based on the POS tagging. Specifically, we employ one hot vector in which each tag is transformed into a $K$-dimensional vector. Similar to (*Jebbara & Cimiano, 2016*), we use a Stanford POS Tagger with 45 tags. These are encoded as 45-dimensional vector and represented as a matrix. This can be represented as: $|S|_1^n = \{s_1 \ldots \ldots, s_n\}, \in R^{45}$.

## Convolutional layer

After all the textual information is encoded into vectors and zero padding is applied to make all the embedding channels of the same length, the convolution operations are then applied to generate local features. Thus, the main purpose of the convolutional layer is to extract local features from the embedding layer. Here, we use two different filter sizes for POS feature $P$ and Semantic Feature $Z$ accordingly. Typically, convolution is a dot product involving filters with weights $W \in R^{hk}$ and a vector of h-gram in a sentence (*Kim, 2014*).

Let $w_p \in R^{hk}$ and $w_z \in R^{hK}$ be filter applied to $h$-gram for the matrix $\mathbf{P}$ and matrix $\mathbf{Z}$, respectively. Then the features generated can be given as:

$$C_i = f(w \cdot x_{i+h} + b) \tag{1}$$

Where $f$ is a nonlinear function (such as hyperbolic tangent or ReLU), $b$ stands for a bias term.

This is applied to each window, $[x_{1:h}, x_{2:h+1}, \ldots x_{n-h\,:n,}]$. With the $w_p \in R^{n-k+1}$ and $w_z \in R^{n-k+1}$, for the matrix $P$ and matrix $Z$, respectively. The features generated for $p$ is given by:

$$c_p = [c_1^p, c_2^p \ldots . c_{n-h+1}^p] \tag{2}$$

And similarly, the feature map for matrix $Z$, is given as

$$c_z = [c_1^z, c_2^z \ldots . c_{n-h+1}^z] \tag{3}$$

However, it is worth to mention that, different semantic and POS features can be extracted using several filters.

## Max pooling layer

Pooling operation is basically aimed to reduce the feature resolution maps by applying a pooling function to several units in a local region of a size based on a parameter known as pooling size. The pooling operation generally serves as generalizations over the features captured from the convolutional operation. Thus, the basic idea behind utilizing max poling layer is to extract the most salient features from the convolutional layer. Typically, pooling layer takes the maximum element in each generated feature map. This can be given as: $\check{C}_p = \max[c_1^p, c_2^p \ldots . c_{n-h+1}^p]$ and $\check{C}_z = \max[c_1^z, c_2^z \ldots . c_{n-h+1}^p]$ for $\mathbf{P}$ and $\mathbf{Z}$, respectively.

When the max pooling is applied, the final maximum feature is generated by concatenating the semantic and POS features using a filter. This can be given as $C = \check{C}_p \oplus \check{C}_z$. Where $\oplus$ is the concatenation operator. As we use several features for the POS and semantic features, we have the final feature as:

$$C = \check{C}_p^1 \oplus \ldots .. \oplus \check{C}_p^n \oplus \check{C}_z^m \oplus \ldots .. \oplus \check{C}_z^m \tag{4}$$

Where $\boldsymbol{n}$ and $\boldsymbol{m}$ are the filters for semantic and POS features, respectively.

## Output layer

Here, we finally apply the Softmax classifier to generate the probability distribution over given aspects. The main idea of the Softmax function is to carry out a classification process over the high-level features generated from the convolution operation and pooling layers. In this case, the Softmax function is used to find the probability distribution for all the output labels. Here, we specifically treat the aspect extraction as a sequence labeling process. Particularly we apply IOB scheme to indicate our aspect annotations as a tag

sequence. Each word in the text is assigned with one of the three tags: I, O or B indicating beginning, Inside or Outside of an aspect term, respectively.

## MODEL VARIATIONS

In order to evaluate our model, we conducted a series of experiments with different settings of the model.

- MCNN-Random: To assess the impacts of word embeddings, the word embedding channel is randomly initialized while the input channel containing the POS Tag embeddings is ignored, meaning that only the randomized word embeddings channel is considered for training.
- MCNN+W2V: Here, the word embedding layer is initialized with a pretrained word2vec and optimized during training. In particular we used general purpose word embeddings trained on the Google corpus (*Mikolov, Yih & Zweig, 2013*).
- MCNN+W2V2: This is similar to the MCNN+W2V setting; however, instead of using the general pretrained word embedding, we used a domain-specific word2vec trained on either Amazon or Yelp review datasets. This is specifically aimed to assess the impacts of the domain-specific word embeddings compared to the general word embeddings.
- MCNN+W2V+POS: In this case, all the two input channels are considered for the training and optimization process. Specifically, we used the general word embeddings in one channel and POS Tag embeddings in the other channel. However, the model parameters were fine-tuned during optimization
- MCNN+W2V2+POS: This is similar to MCNN-W2V+POS variant; however, in this case, instead of applying a general pretrained word2vec, a domain-specific word embedding was used. All the parameters were fine-tuned.

## EXPERIMENTAL STUDY

In this section, we first present a description of the datasets used, then provide a detailed experimental procedure for evaluating the performance of the proposed approach and finally make a comparison against the baseline methods. We use Recall, Precision and F1 score as the evaluation metrics to evaluate the performance of the model. These metrics have been previously used in several relevant works (*Poria, Cambria & Gelbukh, 2016*; *Popescu & Etzioni, 2005*).

### Dataset

To evaluate the performance of the model, we utilized four different benchmark datasets. The datasets which comprise training and test snippets were collected manually and made available by the organizers for the SemEval competitions. The first two datasets are from SemEval2014 (*Pontiki & Pavlopoulos, 2014*) comprising reviews from laptop and restaurant domains, respectively, while the third and fourth datasets are from semeval2015 (*Maria et al., 2015*) and SemEval2016 (*Hercig et al., 2016*), respectively containing reviews from restaurant domain. These datasets comprise review sentences with aspect

**Table 1 SemEval challenge datasets showing the number of sentences and the aspect terms.**

| Datasets | Train | | Test | |
|---|---|---|---|---|
| | Sentence | Aspect | Sentence | Aspect |
| SemEval2014-L | 3041 | 2358 | 800 | 654 |
| SemEval2014-R | 3045 | 3693 | 800 | 1134 |
| SemEval2015-R | 1315 | 1192 | 685 | 678 |
| SemEval2016-R | 2000 | 1743 | 676 | 622 |

**Note:**
L and R represent the laptop and restaurant domains, respectively.

**Table 2 Examples of aspect and aspect terms word distribution in the laptop and restaurant domains.**

| Domain | Aspect | Aspect terms |
|---|---|---|
| Laptop | Price | Price, regret, deal, money, store, stars, gift, penny, worth |
| | Warranty | Warranty, center, policy, support, repair, service, extended, longer, contact |
| | Design | Exterior, wheels, plastic, wheel, design, interior, wheels, clean, good |
| Restaurant | Service | Manager, owner, staff, workers, employees, messenger, chefs, cleaner |
| | Food | Crispy, tender, chicken, beef, shrimp, curry, tuna, egg, onions |
| | Ambience | Setting, décor, lighting, wall, elegant, cool, nice, trendy |

terms labelled as spans of characters. Tables 1 and 2 show the statistics of the datasets and a typical example of the aspect terms distribution of the laptop and restaurant domains, respectively.

In order to initialize the word vectors, we particularly exploit two different word embeddings: (1) General embeddings in which we use pretrained Google word2vec trained on 100 billion words of google news corpus (*Mikolov, Yih & Zweig, 2013*) using CBOW architecture; (2) Domain-specific embeddings trained on the restaurant review from the Yelp datasets, and electronics reviews from the Amazon datasets, for the restaurant and the laptop domains, respectively. The Yelp (https://www.yelp.com/dataset/) and Amazon (*McAuley & Leskovec, 2013*) review datasets contain 2.2 million and 142.8 million reviews, respectively. We use Gensim which has the implementation of CBOW to train all the datasets. Words that appear less than five times in the review are replaced with <other> token.

## Preprocessing

We carry out preprocessing with the aim of obtaining a clean and structured textual review. Specifically, we convert all the reviews into lower case comprised of only English texts and split the text into separate sentences. We apply noise removal strategy which includes removal of words with special characters, stop words, alphanumeric characters and words that have a length less than or equal to 1.

## Experimental setup

We use fivefolds cross-validation strategy to choose the hyperparameters. Specifically, we choose three filter size of (3, 4, 5), with 100 feature maps. We used a max pooling layer

after each convolutional layer. As we wanted to tag each word, we use 1 as the stride for each convolutional layer. To tackle the issue of the parameter overfitting, we utilized drop out regularization on the penultimate layer with $L2$ constraints of 3. The training is conducted using stochastic gradient descendent over shuffled mini batches of size 64 and a dropout rate of 0.5. We apply ReLU for all the datasets and used 128 to be the size of the hidden rate. These values were chosen based on the careful grid search on the validation subset.

To better assess the performance of the proposed model, we first identify the best performing settings of the model (as described in section 3E) and then make a further comparison with the following baselines models:

- DLIREC (*Toh & Wang, 2015*): The winning system in the SemEval2014 competition (subtask 1) which employ a variety of lexical and semantic features derived from NLP source to improve the performance of the model.
- IHSR & D (*Chernyshevich, 2015*): Another top winning system in the semeval2014 competition which typically exploits CRF and additional features including lexical and statistical features for improving the performance.
- NLANGP (*Toh & Su, 2016*): The top system for semeval2016 challenge for the restaurant domain.
- WDEmb (*Yin et al., 2016*): A dependency-based approach integrated CRF with path embedding for aspect term extraction.
- RNCRF (*Wang et al., 2016*): This model jointly uses CRF and a dependency-based recursive neural network for coextracting aspects and opinion terms. The method also exploits additional handcrafted features.
- CMLA (*Wang, Pan & Dahlmeier, 2017*): A multilayer coupled-attention model for opinion and aspect terms coextraction.
- MIN (*Li & Lam, 2018*): A multitask learning approach that exploits lexicons and dependency rules to jointly perform coextraction of aspect terms and opinion terms. It uses two different LSTMs for the polarity classification of sentences.
- DTBCSNN (*Ye, Yan & Luo, 2017*): A dependency tree based convolutional stacked neural network which used the inference layer for aspect extraction.
- DE-CNN (*Xu et al., 2018b*): A CNN based model exploiting double embeddings for aspect extraction.
- BiDTreeCRF (*Luo et al., 2018*): A tree-based deep learning approach which uses bidirectional LSTM and the CRF layer for improving aspect extraction.

## RESULTS AND DISCUSSION

Table 3 shows the results of the proposed model compared to the baseline models. Here, the results of the best two settings of the model were recorded for each dataset. It can be shown that the best performing variants of the proposed model significantly outperform the state of art approaches. The statistical $t$-test shows the improvement is significant at the confidence level of 95%.

**Table 3  Comparison results of our best performing model variants in terms of F1 scores (%) with the state-of-the-art methods.**

| Model | SemEval2014-L | SemEval2014-R | SemEval2015-R | SemEval2016-R |
|-------|---------------|---------------|---------------|---------------|
| HIS_RD | 74.55 | 79.62 | – | – |
| NLANGP | – | – | 67.12 | 72.34 |
| DLIREC | 73.78 | 84.01 | – | – |
| WDEmb | 75.16 | 84.97 | 69.73 | – |
| RNCRF+F | 78.42 | 84.93 | – | – |
| CMLA | 77.8 | – | – | – |
| MIN | 77.58 | 85.29 | 70.73 | 73.44 |
| BidTreeCRF | 80.57 | 84.83 | 70.83 | 74.49 |
| DTBCSNN | 75.66 | 83.97 | | |
| DE-CNN | **81.59** | – | – | 74.37 |
| **MCNN+WV+POS** | 79.84 | 84.69 | **72.84** | 72.62 |
| **MCNN+WV2+POS** | 80.63 | **86.89** | 72.65 | **75.71** |

**Note:**
Values in bold represent best results.

Compared to the best-performing systems in the SemEval competitions, our model performs better than HIS_RD and DLIREC with gains of 6.08%, 7.27% and 6.85%, 2.88% F1 score on the semEval2014-L and SemEval2014-R datasets, respectively. Similarly, our approach also achieves significant gains against NLANGP by 5.72% and 3.37% F1 score on the SemEval2005-R and SemEval2016-R, respectively. It can be observed that even the WDemb approach which exploits word dependency with additional embedding, still our model achieved significant gains compared to the model across all the datasets. One can also notice from Table 3 that, our model outperforms MIN which is a multitasking approach, with a gain of 3.05%, 1.6%, 2.11% and 2.27% F1 score on the SemEval2014-L, SemEval2014-R, SemEval2015-R and SemEval2016-R datasets, respectively. Our model also outperforms CMLA which is a multilayer approach by 2.83% F1 score on the semeval2014-L datasets.

Despite exploiting the additional handcrafted features by RNCR+F and DTBCSNN, still, our approach achieves 2.21%, 1.96% and 4.97%, 2.92% F1 score gains over the two approaches on the semeval2014-L and semeval-2014-R datasets, respectively. Moreover, our model outperforms the recent tree-base bidirectional method, BidTreeCRF by 0.06%, 2.06%, 2.01% and 1.22% F1 score on the semeval2014-L, semeval2014-R, semeval2015-R and semeval2016-R datasets, respectively. Compared to the double embedding CNN approach, DE-CNN which is the state-of-the-art double embedding approach, our model suffered a low performance on the semeval2014-L, however, it manages to achieve a gain of 1.34% F1 score on the semeval2016-R datasets. This apparently shows the superior performance of our model over the DE-CNN model.

It can be observed from Table 4, that different settings of the model have different performances across the four different datasets. MCNN-WV2-POS performs better than all the other variants across all the datasets while the MCNN-random records relatively lowest performance except on the semeval2015-R where the MCNN-WV2-POS records

**Table 4 Comparison results of the different variants of our model in terms of recall, precision and F1 score (%) performance.**

| Variant | SemEval2014-L | | | SemEval2014-R | | | SemEval2015-R | | | SemEval2016-R | | |
|---|---|---|---|---|---|---|---|---|---|---|---|---|
| | R | P | F | R | P | F | R | P | F | R | P | F |
| MCNN+Rand | 68.50 | 73.41 | 70.87 | 80.76 | 83.45 | 82.08 | 60.20 | 70.50 | 64.94 | 65.61 | 70.25 | 67.85 |
| MCNN+WV | 74.30 | 82.65 | 78.25 | 83.50 | 85.20 | 84.34 | 62.60 | 73.01 | 67.41 | 68.71 | 74.32 | 71.40 |
| MCNN+WV2 | 75.85 | 86.61 | 80.87 | 85.71 | 86.14 | 85.92 | 65.54 | **75.87** | 70.33 | 70.56 | 74.54 | 72.50 |
| MCNN+WV+POS | 74.85 | 85.54 | 79.84 | 83.32 | 86.10 | 84.69 | **71.32** | 74.43 | **72.84** | 69.12 | 76.50 | 72.62 |
| MCNN+WV2+POS | **77.65** | **86.65** | **81.90** | **86.24** | **87.01** | **86.62** | 70.08 | 75.41 | 72.65 | **72.17** | **79.61** | **75.71** |

**Note:**
  Values in bold represent best results.

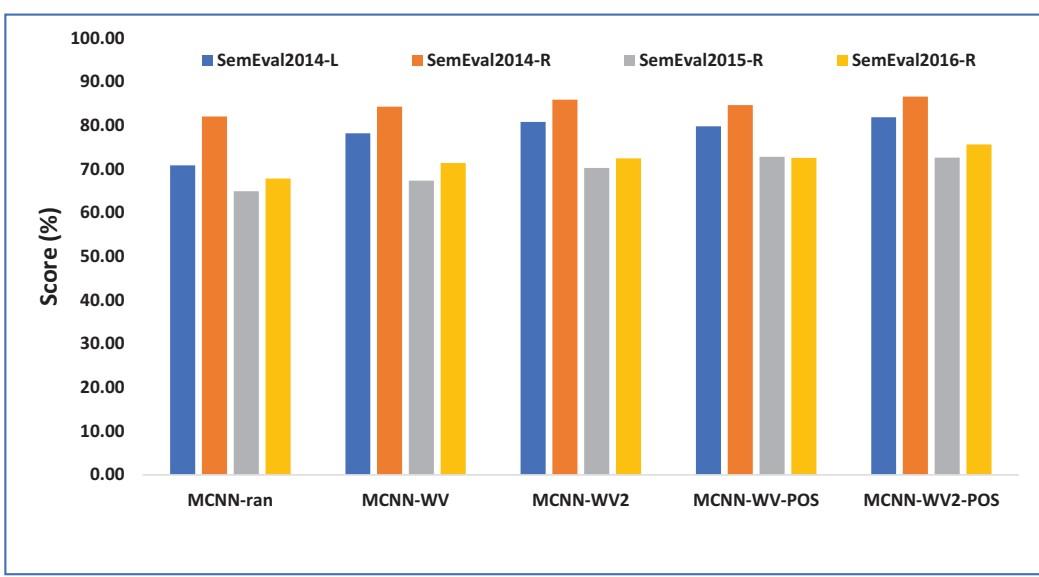

**Figure 2 Performance of our model variants in terms of F1 score accuracy.** Each point indicates an F1 score performance computed in percentage (%).

the best results. This is likely due to the relatively smaller size of the semeval2015-R datasets. Similarly, one can notice from the Table 4, that in all the variants, the best results were recorded on the restaurant domain while relatively lower results are recorded on the laptop domain in all the datasets. This is likely due to the lower number of the aspects term contained in the restaurant domain than in the laptop review domain.

As can be seen from Table 4 and Fig. 2, all the variants of our model with the exception of MCNN-random demonstrate relatively competitive results with significant improvement across all the domains. This specifically indicates the weakness of the randomly initialized word embeddings for the aspect extraction. This is because MCNN-random is randomly initialized while the other variants are particularly initialized with pretrained word embeddings. This translates the importance of pretrained word embeddings over the randomly initialized word embeddings. The results also show that using domain-specific word embeddings for both laptop and restaurant domains perform better than the general word embeddings (Google embeddings) initialization. This

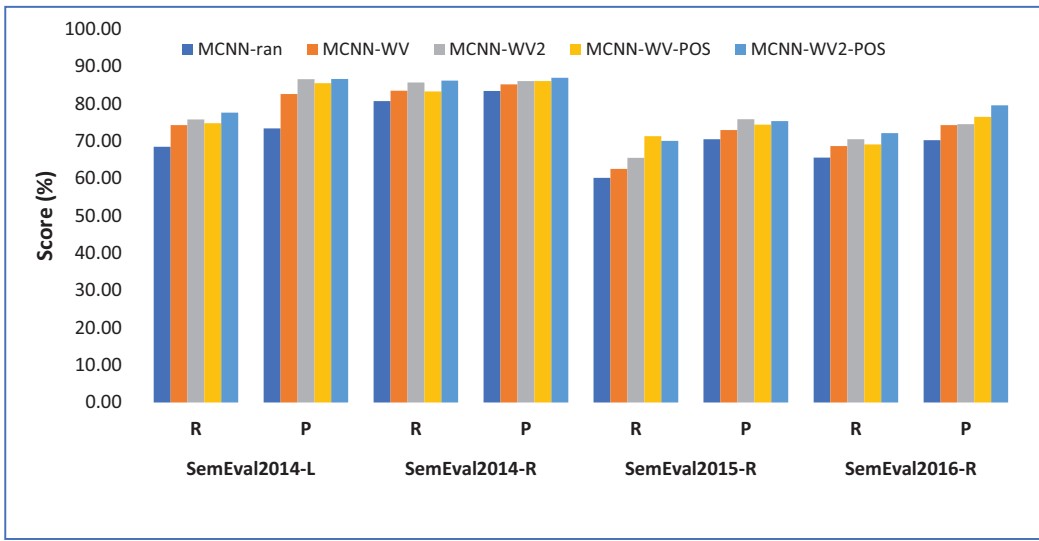

**Figure 3 Performance of the different model variants in terms of recall and precision.** Each point shows precision and recall performance measured in percentage (%).

supports the intuition that domain-specific embeddings typically contain opinion specific information related to a particular domain (laptop or restaurant) which helps to perform better than the general word embeddings which are merely trained on the Google News corpus which is typically composed of textual reviews about the commonly discussed matters on the news.

One can observe from Fig. 3 that in both laptop and restaurant domain the model suffers from low recall, meaning that it missed some vital aspect terms. However, using the POS tag which is an important linguistic feature help to overcome some drawbacks thereby improving the performance of the model. This specifically indicates the importance of using POS tags features in addition to pretrained word embeddings in aspect term extraction.

We further conduct an experiment to assess the sensitivity of the model towards word embeddings dimensions. We specifically use different word embedding dimensions from 50 to 375 with the intervals of 25, i.e., {50, 75, 100, 125, 150, 175, 200, 225, 250, 275, 300, 325, 350, 375}. The laptop domain uses embeddings trained on the Amazon reviews and restaurant domain use the embeddings trained on the Yelp reviews datasets. Figure 4 shows the experimental results on the MCNN-WV2 variant. The results indicate the highest performance at around 300 dimensions and relatively remains stable above 150. This particularly implies the insensitivity of the model toward the dimension of word embeddings provided it is within the appropriate range such as 100–375.

It is clear that two key factors are basically the reasons behind the good performance of our model compared to the baseline methods: First, the POS tag embedding input layer which helps for better sequence labeling. The domain-specific pretrained word embeddings which were trained on the target domain corpus of the review datasets. The advantage of our approach is that it is relatively uncomplicated and automatic that

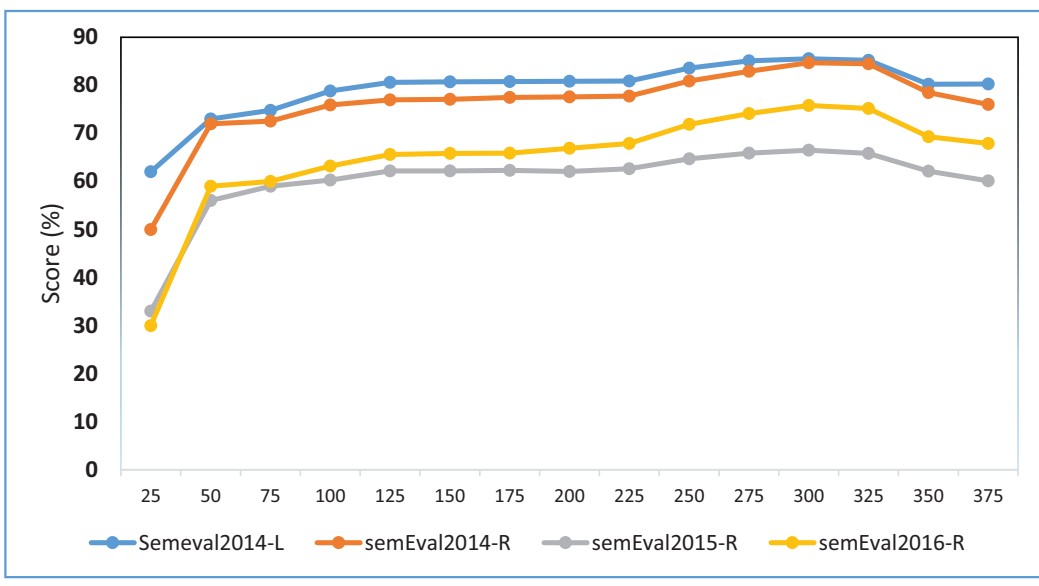

**Figure 4 F1 score of the MCNN-WV2-POS Variant of our model on different word embedding dimensions.** Each point shows the F1 performance measured in percentage (%).

does not require any feature engineering. This saves time, cost and improves the high performance of the model.

## CONCLUSION

In this research, we proposed an aspect extraction approach using Deep MCNN leveraging two different channels namely, word embeddings and POS tag embeddings. We presented a series of experiments and the results on various datasets showed that our proposed approach outperformed the state-of-the-art methods. Our results support the previous findings that showed that pretrained word vectors are always better than randomly initialized embeddings in deep learning based NLP tasks such as aspect extraction. It also reaffirms many of the previous findings which show that exploiting POS tag features improves the performance of NLP methods including aspect extraction. We also demonstrated the importance of using a domain specific word embedding over the general word embeddings. As a future direction of research, we think that applying an attention-based deep learning model for improving aspect extraction is worth exploring, and that integrating a lexicon in the word embedding layer in the MCNN is another direction of further exploration.

### Funding

The authors received no funding for this work.

### Competing Interests

The authors declare that they have no competing interests.

## Author Contributions

- Aminu Da'u conceived and designed the experiments, performed the experiments, analyzed the data, contributed reagents/materials/analysis tools, prepared figures and/or tables, performed the computation work, authored or reviewed drafts of the paper.
- Naomie Salim conceived and designed the experiments, analyzed the data, contributed reagents/materials/analysis tools, performed the computation work, authored or reviewed drafts of the paper, approved the final draft, proof reading.

## Data Availability

The relevant datasets and the snippets codes are available in the Supplemental Files.

## Supplemental Information

Supplemental information for this article can be found online at http://dx.doi.org/10.7717/peerj-cs.191#supplemental-information.

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
