# Peer review of "Aspect extraction on user textual reviews using multi-channel convolutional neural network"

_PeerJ Computer Science, doi:10.7717/peerj-cs.191_

## Round 0.1 · original submission · Major Revisions

Please address all the referees' comments.

Reviewer 1 ·

Basic reporting

see below

Experimental design

see below

Validity of the findings

see below

Additional comments

The manuscript is centered on an interesting topic.
Organization of the paper is good and the proposed method is quite novel.

The manuscript, however, does not link well with recent literature on sentiment analysis appeared in relevant top-tier journals, e.g., the IEEE Intelligent Systems department on "Affective Computing and Sentiment Analysis". Also, check latest trends in deep learning based aspect extraction, e.g., attentive LSTM.

Many issues with acronyms: some acronyms are used without being defined first, e.g., CNN, some other are defined but then never used, e.g., CRF.

Some bad English constructions and misuse of articles: a professional language editing service is strongly recommended (e.g., the ones recommended by IEEE, Elsevier, and Springer) to sufficiently improve the paper's presentation quality for meeting PeerJ's high standards.

Reviewer 2 ·

Basic reporting

• The article is generally written in a clear, coherent and professional English. However, there are some segments that need to be corrected because they are not well understood or they are incorrectly written. For the next revision, it is necessary to improve the segments at lines: 13, 38, 40, 88 to 91, 92, 95 to 97, 153, 191 to 192, 262, 320 to 321, 396 to 397, and 476.
• The Introduction and background are correct because they reveal the problems that need to be addressed and clearly explain the objectives of the paper. And the related works are generally well-founded because they explain clearly the works that have addressed the problem of aspect extraction.
• The work is very well structured.
• The tables and figures are relevant and help to better understand the methodology and results obtained, however, figures 2 and 3 need to be edited and eliminate the title left by mistake in both figures: “Chart Title”. In addition, it would be good to explain a little better all the figures in the caption that corresponds to them.
• Conclusion of basic reporting: Approved

Experimental design

• The research of the article satisfies the Aims and Scope of the journal
• The problems existing with the aspect extraction of reviews task and how these problems can be solved with the paper proposal are clearly specified. In addition, the suggested methods are quite relevant and novel for the current literature. As a suggestion, the research question should be clearly specified so that it can then be answered in the conclusions of the work.
• Many works found in the literature have been explained, which is quite good. However, it would be ideal to explain a little more the advantages and disadvantages of those works that you consider most important and that are better related to the proposal of your work, that is, you might detail a little more explaining their performance.
• The methodology of the work is clearly explained, the methods are described in detail.
• In the part of the explanation of the datasets, it would be interesting to show some examples of the most important aspects that are found in each dataset, no example was shown. Explain a little better the datasets used.
• Conclusion of experimental design: Approved

Validity of the findings

• The article showed a large number of experiments with relevant results. The data is robust and statistically sound and meets the objectives that were raised in the introduction of the article.
• The conclusions are well established and explain the importance of the results that have been obtained. However, since the research question has not been clearly defined previously (in the introduction), the conclusion seems a bit incomplete. The research question should be defined and finally answer that question according to the results obtained in this work.
• Conclusion of validity of the findings: Approved

Reviewer 3 ·

Basic reporting

The manuscript entitled "Aspect extraction on user textual reviews using Multi-Channel Convolutional Neural Network" presents an approach for aspect extraction using deep multichannel convolutional neural network. The main topic is relevant and interesting to the scope of this journal. I would like to highlight some points about the general organization of the paper that should be revised by the authors.

- I recommend that the manuscript be reviewed for what concerns the English language for both the composition of the sentences and the grammar.
- The text also needs more attention with some sentences. For instance, "As a feature direction of the research" (line 480)

- There are many abbreviations in the introduction that were not defined in the text, like "LSTM".
- "CRF" is defined twice.

- Some figures need to be improved. In Fig. 2 it is missing the legend for the orange bar. In Fig. 3, the x-axis is not shown.

The literature references and the related works are well structured and the main background of the manuscript is provided.

Experimental design

The main theoretical tools approached by the authors are cutting-edge research on many machine learning and related topics nowadays and the authors explore their architecture for aspect extraction within the context of sentiment analysis. I missed some motivation for what concerns this main application of the method. I think that the authors could improve the text by adding some real-world examples of the datasets that were used. This way, they can illustrate the main application of the proposed approach and motivate the reader to go further in the methodology and the results.

Validity of the findings

The research question is well defined in the manuscript. The authors performed a deep investigation of the results by making comparisons with other methods in literature and also with some variations of the proposed model. The authors showed that the proposed approach could outperform many state of the art methods. Therefore, the main contribution of the manuscript is the performance gain obtained by the proposed method.

---

## Round 0.2 · Minor Revisions

The authors have addressed several issues, however the referees pointed out that some issues still remain. Please address all referees' comments.

Reviewer 1 ·

Basic reporting

All my comments have been addressed. However, the compilation of some references should be double-checked, e.g., see "Opinion Miningwith"

Experimental design

See my previous review

Validity of the findings

See my previous review

Reviewer 2 ·

Basic reporting

All the comments that were suggested in the previous revision have been addressed. However the captions of Figure 4 and some Tables presents some errors that need to be reviewed:
Figure 4: F1score
Table 1: Rrepresent
Table 2: Exampleof worddistribution
Table 3: interms
Table 4: Comparisonresults

Experimental design

no comment

Validity of the findings

no comment

Additional comments

The revisions are satisfactory, except that some minor errors need to be reviewed (see basic reporting).

Reviewer 3 ·

Basic reporting

no comments

Experimental design

no comments

Validity of the findings

no comments

Additional comments

The manuscript was greatly improved in this revised version. The main points raised by the reviewers were addressed by the authors. However, I have some minor comments in order to make the manuscript clearer:

The names of the datatsets do not need to be mentioned in the abstract (line 32) unless they would be recognized just by their names.

The second paragraph of section "Related Work" of the tracked changes version is different from the final revised version. The same thing happend in Section 3.1. The sentence starting with "Similar to [27] each word in the sentence... " in the revised version is not present in the final version. The authors should check if both versions are the same.

Line 94: a reference is missing for the SemEval challenge.

Line 114 should be corrected as "For instance, 'Author et al.' [17] employed RMB....". The authors should check other places in the manuscript with similar notation, mainly in the Related Work section. It is also weird to start a sentence with "[n]" as the example of line 115. I recommend using the name of the authors instead or rearranging the sentence.

Still, some figures are imcomplete. In Figs. 1, 2 and 3, the y-axis title is missing. It should refer to performance. In addition, in Fig. 3, the x-axis is not there.

Line 147: "...is made up of..."

Caption of Table 2: "word distribution"

Caption of Table 4: "Comparison results"

Line 306: "... baseline models"

---

## Round 0.3 · Minor Revisions

We noted that in some parts your text should be improved.
As we don’t offer language editing when articles are in production, it is definitely recommended that you use an editing service to check the language.

Reviewer 2 ·

Basic reporting

no comment

Experimental design

no comment

Validity of the findings

no comment

Additional comments

All comments have been addressed.

Reviewer 3 ·

Basic reporting

no comments

Experimental design

no comments

Validity of the findings

no comments

Additional comments

All the points raised by the reviewers were addressed by the authors. Now the text and figures are clearer and ready for publication.

---

## Round 0.4 · accepted · Accept

Your article is now Accepted.

Reviewer 1 ·

Basic reporting

can't comment because I can't see previous issues

Experimental design

can't comment because I can't see previous issues

Validity of the findings

can't comment because I can't see previous issues

Additional comments

please write a proper rebuttal